# Impact of Comorbidities, Sarcopenia, and Nutritional Status on the Long-Term Outcomes after Endoscopic Submucosal Dissection for Early Gastric Cancer in Elderly Patients Aged ≥ 80 Years

**DOI:** 10.3390/cancers13143598

**Published:** 2021-07-18

**Authors:** Ga Hee Kim, Kee Don Choi, Yousun Ko, Taeyong Park, Kyung Won Kim, Seo Young Park, Hee Kyong Na, Ji Yong Ahn, Jeong Hoon Lee, Kee Wook Jung, Do Hoon Kim, Ho June Song, Gin Hyug Lee, Hwoon-Yong Jung

**Affiliations:** 1Asan Medical Center, Department of Gastroenterology, University of Ulsan College of Medicine, Seoul 05505, Korea; smallgh@hanmail.net (G.H.K.); hkna77@naver.com (H.K.N.); ji110@hanmail.net (J.Y.A.); estampie@gmail.com (J.H.L.); jung.keewook30@gmail.com (K.W.J.); dohoon.md@gmail.com (D.H.K.); hjsong@amc.seoul.kr (H.J.S.); jhlee409@amc.seoul.kr (G.H.L.); hwoonymd@gmail.com (H.-Y.J.); 2Biomedical Reseach Center, Asan Institute for Life Sciences, University of Ulsan College of Medicine, Seoul 05505, Korea; ko.yousun82@gmail.com; 3Asan Medical Center, Department of Radiology, University of Ulsan College of Medicine, Seoul 05505, Korea; pak14kr@gmail.com (T.P.); medimash@gmail.com (K.W.K.); 4Department of Statistics and Data Science, Korea National Open University, Seoul 05505, Korea; biostat81@gmail.com

**Keywords:** early gastric cancer, elderly, endoscopic submucosal dissection, prognostic nutritional index, Charlson comorbidity index

## Abstract

**Simple Summary:**

The average human life expectancy is increasing worldwide, which is leading to increases in the proportion of elderly patients (≥80 years) with gastric cancer. Endoscopic submucosal dissection (ESD) is widely regarded as a safe and effective treatment for early gastric cancer (EGC), even in aged or high-risk patients. We investigated the oncologic outcomes in elderly patients who underwent endoscopic submucosal dissection (ESD) for early gastric cancer (EGC) by focusing on the impact of comorbidities, sarcopenia, and nutritional status. Over a median follow-up period of 70.5 months, the 3- and 5-year overall survival (OS) rates were 89.5% and 77.1%, respectively; of the114 patients who died, only four (3.5%) were due to gastric cancer. A total of 173 (61.8%) had sarcopenia, and they had lower rates of 3-year (88.4% vs. 91.4%) and 5-year (73.1% vs. 84.0%; *p* = 0.046) OS than did those without sarcopenia. In multivariable analyses, prognostic nutritional index (hazard ratio [HR], 0.93; 95% confidence interval [CI]: 0.90–0.98; *p* = 0.002) and Charlson comorbidity index (HR 1.19; 95% CI: 1.03–1.37; *p* = 0.018) showed that ESD was a feasible and safe therapeutic method to use in elderly patients, whose long-term survival was significantly associated with nutritional status and comorbidities.

**Abstract:**

Background/Aim: We investigated the oncologic outcomes in elderly patients who underwent endoscopic submucosal dissection (ESD) for early gastric cancer (EGC) by focusing on the impact of comorbidities, sarcopenia, and nutritional status. Methods: Between 2005 and 2016, 280 patients aged ≥ 80 years with 289 EGCs underwent ESD at a tertiary care center. The short- and long-term survival outcomes were assessed. Cox regression analysis was used to identify factors associated with survival, including clinicopathologic factors and abdominal muscle area measured by computed tomography. Results: The rates of *en bloc*, R0, and, curative resection were 99.3%, 90.0%, and 69.2%, respectively. The rates of post-ESD bleeding and perforation rates were 2.1% and 3.1%, respectively, and no cases showed significant life-threatening adverse events. Over a median follow-up period of 70.5 months, the 3- and 5-year overall survival (OS) rates were 89.5% and 77.1%, respectively; of the114 patients who died, only four (3.5%) were due to gastric cancer. A total of 173 (61.8%) had sarcopenia, and they had lower rates of 3-year (88.4% vs. 91.4%) and 5-year (73.1% vs. 84.0%; *p* = 0.046) OS than did those without sarcopenia. In multivariable analyses, prognostic nutritional index (hazard ratio [HR], 0.93; 95% confidence interval [CI]: 0.90–0.98; *p* = 0.002) and Charlson comorbidity index (HR 1.19; 95% CI: 1.03–1.37; *p* = 0.018) were significant factors associated with overall survival. Conclusions: ESD was a feasible and safe therapeutic method to use in elderly patients, whose long-term survival was significantly associated with nutritional status and comorbidities. These results suggest the need for a possible extension of the curative criteria for ESD in elderly patients with EGC.

## 1. Introduction

The average human life expectancy is increasing worldwide, which is leading to increases in the proportion of elderly patients with gastric cancer. Endoscopic submucosal dissection (ESD) is widely regarded as a safe and effective treatment for early gastric cancer (EGC), even in aged or high-risk patients [1,2,3]. Previous studies have reported that Eastern Cooperative Oncology Group (ECOG) performance status [4], Charlson comorbidity index (CCI) [1,5,6], prognostic nutritional index (PNI) [5,7], and American Society of Anesthesiologists (ASA) physical status classification [4] are independent prognostic factors in elderly patients who undergo ESD for EGC. However, there are limited data on the therapeutic strategy for elderly patients with EGC undergoing ESD according to prognostic factors.

Sarcopenia is a geriatric syndrome, characterized by progressive and generalized loss of skeletal muscle mass and strength, and it has been recognized as a significant predictor for morbidity and mortality after operation [8,9]. Although some studies have described the implications of sarcopenia on surgical outcomes and prognosis in patients with gastric cancer, only a few studies have investigated the prognostic impact of sarcopenia in elderly patients undergoing ESD for EGC.

In the present study, we investigated the oncologic outcomes and prognostic factors in elderly patients (≥80 years) who underwent ESD for EGC by focusing on comorbidities, sarcopenia, and nutritional status for elderly patients.

## 2. Methods

### 2.1. Study Populations and Data Collection

All patients aged ≥80 years who were diagnosed with EGC and underwent ESD between 2005 and 2016 at Asan Medical Center (Seoul, Korea) were retrospectively included in this study. The clinicopathological characteristics and treatment outcomes of the patients were reviewed using electronic medical records. Baseline data on age, sex, ECOG, comorbidity, CCI, ASA physical status classification, and tumor characteristics (i.e., size, location, size, morphology, appearance, histology, and depth of invasion) were collected. This study was approved by the institutional review board of Asan Medical Center (approval no. 2020–0584).

### 2.2. Endoscopic Submucosal Dissection

All ESD procedures were performed by experienced endoscopists (K.D.C., J.Y.A., D.H.K., J.H.L., S.H.J., L.G.H., and H.Y.J.) using a single-channel endoscope (GIF-H260 or GIF-HQ290; Olympus, Tokyo, Japan), with various knives, including insulation-tipped (IT), IT-2, hook, and dual knives. Procedures were performed with the patient under sedation using intravenous injection of midazolam (0.05 mg/kg) with pethidine (50 mg). The typical ESD procedures included marking, mucosal incision, and submucosal dissection. After dissection, preemptive endoscopic hemostasis was performed for oozing or exposed vessels. The procedure time was defined as the time from the completion of marking to the removal of the scope. Perforation was defined as the visualization of a breach in the muscularis propria at endoscopy, with fluid in the peritoneum, with or without clinical symptoms. Postoperative bleeding was determined by the presence of hematemesis, tarry stool, or a decreased hemoglobin level after completion of the ESD procedures and removal of the endoscope. Second-look endoscopies were routinely performed on the second day after the procedure. When active bleeding or large blood clots were discovered during procedure, endoscopic hemostasis was performed.

### 2.3. Histopathologic Evaluation

The histological type of the tumors was classified according to the World Health Organization classification of tumors. Macroscopically, the tumors were classified as elevated, depressed, or flat. The resected specimens were observed under a stereomicroscope and subsequently cut into 2 mm pieces. Histology was classified into differentiated adenocarcinoma (i.e., well- and moderately differentiated adenocarcinoma and papillary adenocarcinoma) or undifferentiated adenocarcinoma (i.e., poorly differentiated adenocarcinoma or signet-ring-cell carcinoma). Lymphovascular invasion was defined as the observable spread of tumor cells through the lymphatic vessels. The depth of submucosal invasion was categorized into SM1 (<500 μm) or SM2 (≥500 μm) based on the distance from the lowest portion of the muscularis mucosae.

### 2.4. Definitions

Curative resection for EGCs with an absolute indication was performed when all of the following conditions were met: *en bloc* resection, negative horizontal and vertical margins, no lymphovascular invasion, and differentiated intramucosal tumor ≤2 cm without ulcer. Curative resection for EGCs with an expanded indication was performed when a lesion was (a) >2 cm in diameter, predominantly differentiated type, pT1a, and nonulcerated; (b) ≤3 cm, predominantly differentiated type, pT1a, and ulcerated; (c) ≤2 cm, predominantly undifferentiated type, pT1a, and nonulcerated; or (d) ≤3 cm, predominantly differentiated type, pT1b (SM1) with *en bloc* resection, negative horizontal and vertical margins, and no lymphovascular invasion. When the tumors did not fulfill the criteria for curative resection, the resection was considered to be a non-curative resection. The indication of ESD was based on the Korean Gastric Cancer Association guidelines [10]. Tumors detected at the resection site were defined as locally recurrent tumors. Tumors detected at sites other than the primary resection site within 12 months of ESD were regarded as synchronous tumors, and those tumors detected more than 12 months after ESD were regarded as metachronous. Extragastric recurrence was defined as regional recurrence in perigastric lymph nodes and distant recurrence in the liver, lung, bone, brain, distant lymph nodes, and peritoneum, irrespective of intragastric lesions. PNI was calculated as 10 × albumin (g/dL) + 0.005 × total lymphocyte count (per μL).

### 2.5. Measurement of Body Composition

All CT images were retrieved from the Picture Archiving and Communication System at Asan medical center. The presence of sarcopenia was evaluated on abdominal CT with an artificial intelligence software (AID-U^TM^, iAID Inc, Seoul, Korea) that was developed using a fully convolutional network segmentation technique [11]. Experienced operators (Y.K. and K.W.K.), who were blinded to the clinical information, selected the axial CT image at the L3 lumbar vertebra inferior endplate level in a semi-automatic manner with the aid of coronal reconstructed images. Using AID-U^TM^, the selected CT images were automatically segmented to generate the boundary of total abdominal muscles and measure the abdominal muscle and fat area. Then, two operators (Y.K. and K.W.K.) checked the quality of the muscle segmentation in all images. The skeletal muscle area (SMA, cm^2^), which includes all muscles on the selected axial images, i.e., psoas, paraspinals, transversus abdominis, rectus abdominis, quadratus lumborum, internal obliques, and external obliques, were demarcated using the predetermined thresholds of −29 to +150 Hounsfield units on CT (Figure 1). The visceral fat area (VFA, cm^2^) and subcutaneous fat area (SFA, cm^2^) were demarcated using the adipose tissue thresholds of−190 to −30 Hounsfield units on CT. Visceral obesity was defined as a VFA ≥ 100 cm^2^. The skeletal muscle index (SMI) was calculated as SMA/height^2^ and sarcopenia was defined by an SMI of 52.4 cm^2^/m^2^ for men and 38.5 cm^2^/m^2^ for women [12]. Sarcopenic obesity was defined as a VFA/SMA ratio of above 3.2 [13].

### 2.6. Follow-Up Protocol and Outcome Assessment 

Patients were followed up with an endoscopic examination and abdominal CT. Both procedures were performed every 6 months for the first 2 years and then annually for the next 3 years. Overall survival (OS) was defined as the time between the initial endoscopic treatment and death from any cause or censoring. Disease-specific survival was defined as the time between the initial endoscopic treatment and death from gastric cancer. The survival status was determined using the medical records and claims data of the Korean National Health Insurance Service. Patients who underwent non-curative resection were referred for additional surgery; three patients with non-curative resection were fully informed about the benefits and risks of additional surgery. If the patients did not want to undergo gastrectomy or had a high risk of surgical complications, they were followed up without operation.

### 2.7. Statistical Analysis

OS was estimated by the Kaplan–Meier method and compared with the log-rank test. In Cox proportional hazards regression analysis, hazard ratios (HR) and 95% confidence interval (CI) were calculated to investigate the relationships between OS and the clinical factors. In the case of patients who underwent additional surgery because of non-curative resection, we censored them at the date of additional surgery to remove the effect of the additional surgery on survival.

All analyses were performed using R 3.5.1 (R Foundation for Statistical Computing, Vienna, Austria) and *p*-values < 0.05 were considered to be statistically significant.

## 3. Results

### 3.1. Baseline Clinicopathologic Characteristics

A total of 9015 cases of EGC were treated with ESD at Asan Medical Center between 2005 and 2016. Among these, 317 EGCs were in patients aged ≥80 years. After excluding 28 cases (previous gastrectomy or endoscopic resection, *n* = 19; without CT image, *n* = 6; lack of follow-up data, *n* = 3), a total of 280 patients with 289 lesions were analyzed (Figure 2). The median age of the study patients was 82 years (range 80–92), and 182 (65.0%) patients were male (Table 1). The common comorbidities included hypertension (60.7%), diabetes mellitus (17.5%), and cardiac disease (17.5%); 188 (67.1%) patients had a CCI of <2. The mean body mass index was 23.3 ± 3.4 kg/m^2^ and the mean PNI was 46.7 ± 4.8. A total of 173 (61.8%) patients had sarcopenia and 142 (50.7%) patients had visceral obesity.

The characteristics of the 289 lesions are shown in Table 2. The median tumor size was 20 mm (range, 3–85), 263 (91.0%) tumors were differentiated, and 34 (11.8%) tumors had deep submucosal invasion (≥SM2). Lymphovascular invasion was positive in 30 (10.4%) tumors.

### 3.2. Short-Term Outcomes

The short-term outcomes are summarized in Table 3. The rates of *en bloc*, R0, and curative resection were 99.3%, 90.0%, and 69.2%, respectively. The rates of post-ESD bleeding and perforation were 5.9% and 2.1%, respectively; all perforations were per procedural perforation. There were no significant life-threatening adverse events in the study population. Pneumonia and stricture occurred in one (0.3%) and four (1.4%) of cases after ESD, respectively.

A total of 89 patients underwent non-curative resection, of whom additional gastrectomy was performed in six (6.7%) patients. The most common cause of non-curative resection was deep submucosal invasion (37.1%); lymphovascular invasion was positive in 30 (33.7%) tumors, and positive lateral margin and deep margin were observed in 16 (18.0%) and 13 (14.6%) tumors, respectively. Among the 13 (13/89, 14.6%) cases with vertical margin, 8 (8/13, 62%) cases revealed positive margin that overlapped deep submucosal invasion.

### 3.3. Long-Term Outcomes

The median follow-up period was 70.5 months (range, 4–174), during which 114 (40.7%) of the study patients died (Table 4). The 3- and 5-year OS rates were 89.5% and 77.1%, respectively. The 5-year OS rate was significantly lower in patients with sarcopenia (68.5%) than in those without sarcopenia (84.1%; *p* = 0.046; Figure 3a). In contrast, the 5-year OS rate was not significantly different between those who underwent curative ESD and those who underwent non-curative ESD (*p* = 0.93; Figure 3b).

The prognostic factors of OS in elderly patients who underwent ESD for EGC are shown in Table 5. In the univariate analysis, CCI (HR, 1.23; 95% CI, 1.06–1.41; *p* = 0.005), PNI (HR, 0.93; 95% CI, 0.89–0.97; *p* < 0.001), and sarcopenia (HR, 1.51; 95% CI, 1.01–2.27; *p* < 0.048) were significantly associated with OS. In the multivariable analysis, PNI (0.93; 0.90–0.98; *p* = 0.002) and CCI (1.19; 1.03–1.37; *p* = 0.018) were significantly associated with OS. The factors of non-curative resection, such as lymphovascular invasion, positive resection margin, and deep submucosal invasion, were not significantly associated with OS.

One patient was found to have local recurrence after ESD with a positive lateral margin. This patient was treated by additional argon plasma coagulation and redone ESD. However, the patient underwent gastrectomy after the redone ESD because of the positive lateral margin. The patient was alive during the study period after surgery. Extragastric recurrence of primary EGC was found in four (1.4%) patients, and metachronous gastric cancers were observed in 19 (6.8%) patients. During follow-up, four (1.4%) patients died due to the recurrence of gastric cancer; the detailed clinical and endoscopic factors of these patients are summarized in Table 6.

## 4. Discussion

In the present study, we found that the short- and long-term outcomes of ESD in elderly patients with EGC were favorable, despite the fact that more than a quarter of the cases (83/289, 28.7%) were out of indication for ESD. The *en bloc*, R0, and curative resection rates were high (99.3%, 90.0%, and 69.2%) and there were no cases of significant life-threatening adverse events. The elderly population has increased worldwide, and the need for performing ESD in elderly patients with EGC has also increased because surgery is often contraindicated in the elderly due to their poor general condition and high risk of comorbidities. In previous studies, ESD for EGC in elderly patients was reportedly a safe and feasible treatment, even when compared with the results in non-elderly patients [5,14]; although the OS rates were significantly lower in elderly patients, lower than in non-elderly patients, there was no significant difference in the short-term outcomes and local tumor recurrence. In our study, the median survival time was 9.4 years (95% CI 8.3–10.3) after undergoing ESD for EGC. In 2018, the mean life expectancy at 80 years in Korea was 9.3 years [15]. Therefore, it can be inferred that the survival period of the elderly patients who underwent ESD was not shorter than the life expectancy of the general population.

In elderly patients with EGC, both the underlying medical condition and the tumor variables were important factors for OS. In a single-center retrospective study by Sekiguchi et al., low PNI was found to be a prognostic factors in patients aged ≥80 years with EGC who underwent ESD [7]. PNI, which is defined according to the combined parameters of albumin and lymphocytes, may be particularly useful due to its role as a surrogate marker of both inflammation and nutritional status, thus reflecting the presence of both acute inflammation and malnutrition [16]. Accordingly, PNI has been shown to be a prognostic marker of gastrointestinal cancer as well [17,18,19]. CCI, which was developed to assess the risk of death from comorbidities [20], was found to be an independent risk factor for poor OS in elderly patients with EGC treated by ESD [1,5,6]. Accordingly, high CCI (HR, 1.19; 95% CI, 1.03–1.37; *p* = 0.018) and low PNI (HR, 0.93; 95% CI 0.90–0.98; *p* = 0.001) were found to be independent risk factors for OS in our study. These results suggest that comorbidity and nutrition status are important factors when deciding the treatment strategy for elderly patients with EGC.

Sarcopenia can also be used to predict the prognosis of patients with several types of digestive organ cancers, such as esophageal, stomach, pancreas, and colon cancer [18,21,22,23]. In the present study, the 5-year OS rate was significantly lower in patients with sarcopenia (68.5%) than in those without (84.1%; *p* = 0.046). However, in the multivariable analysis, sarcopenia was not an independent predictor of OS (HR, 1.27: 95% CI, 0.84–1.92; *p* = 0.266). Otherwise, PNI (0.93; 0.90–0.98; *p* = 0.001) and CCI (1.19; 1.03–1.37; *p* = 0.018) were significantly associated with OS in the multivariable analysis. This suggests that comorbidities or nutritional status may be more important than sarcopenia in affecting the OS of elderly patients with EGC following ESD.

There has been no prospective study comparing the oncologic outcomes of elderly patients with EGC treated by ESD and those followed up without any treatment. In a recent retrospective study in Japan using propensity matching analysis, the 3-year OS was not significantly different between elderly patients (≥85 years) with EGC who underwent ESD and those who were followed up without any treatment (70.7% vs. 50.3%, *p* = 0.08) [24]. In our study, there was no significant difference in OS between patients with curative ESD and those with non-curative ESD (*p* = 0.93). In total, 89 (30.8%) cases were found to be non-curative. Among them, 29 (32.6%, 29/89) cases underwent ESD while acknowledging that the lesion was out of indication. These patients had poor general condition, high risk of postoperative complication, and expressed preference for ESD over surgery. The decision to perform ESD rather than gastrectomy was discussed with patients and made on an individual basis by taking into account patient preference and comorbidities. Several studies have shown conflicting results. For example, Chang et al. reported that patients aged ≥75 years with curative ESD had a significantly better 5-year OS than those with non-curative ESD (86.9% vs. 72.7%, *p* = 0.037) [6], whereas Sekiguchi et al. reported that there was no significant difference in 5-year OS in patients aged ≥85 years with curative ESD and those with non-curative ESD (69.3% vs. 78.1%, *p* = 0.076) [7]. Likewise, Esaki et al. reported that among patients aged ≥80 years who underwent non-curative ESD, OS was not significantly different between patients with additional surgery and those without (*p* = 0.23) [25].

In our study, 114 (40.7%, 114/280) patients died during the study period. Gastric cancer-related death was reported in only four (3.5%, 4/114) patients with non-curative resection in the follow-up period. The remaining 110 patients (96.5%, 110/114) died from non-gastric cancer causes, such as other malignancy (32.1%, 9 cases), pneumonia (10.3%, 3 cases), and cerebrovascular disease. These findings suggest that it is important to consider factors such as non-curative ESD, as well as comorbidity and PNI, in determining the therapeutic strategy in elderly patients with EGC. Therefore, the results of our study suggest that the decision to perform endoscopic resection in elderly patients should be made with caution when the patient has multiple comorbidities or poor nutritional status.

There are several limitations to this study. Firstly, this was a retrospective, single-center study and may have been affected by selection bias, especially considering that our center is a large-sized tertiary referral hospital. Specifically, we could not find elderly patients with EGC who did not undergo any treatment and could not compare the oncologic outcomes between those who received treatment and those who did not. In addition, because the number of patients who underwent additional gastrectomy after non-curative ESD was small, we could not perform statistical analysis on the effect of additional gastrectomy in patients with non-curative resection. Secondly, the revised definition of sarcopenia in 2018 by the European Working Group on Sarcopenia in Older People 2 consensus is based on low muscle strength, low muscle quality, and low physical performance [26]. However, we could not evaluate the muscle strength and physical performance in the current study. Furthermore, we did not assess the frailty of study patients, which was reported as a potentially important factor on postoperative outcomes [27]. Further studies should use various measurements of sarcopenia and assessment of frailty in elderly patients with EGC. Thirdly, we did not investigate the impact of health-related quality of life in elderly patients after ESD. Despite these limitations, our study is meaningful in that it is the first to analyze the long-term outcomes of elderly patients with EGC who underwent ESD by measuring of body composition with CT images in a large cohort.

## 5. Conclusions

In conclusion, our study found that ESD was a feasible and safe therapeutic method for treating EGC in elderly patients over 80 years of age. Importantly, nutrition status and comorbidity were independent prognostic factors affecting the survival of elderly patients who underwent ESD for EGC. These results suggest the need for a possible extension of the curative criteria for ESD in elderly patients with EGC.

## Figures and Tables

**Figure 1 cancers-13-03598-f001:**
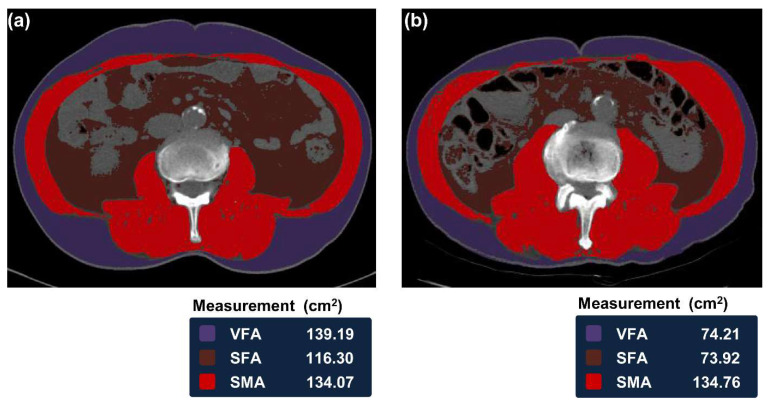
Body morphometric evaluations of the abdominal fat and muscle area of patients with sarcopenia (**a**) and patients without sarcopenia (**b**). VFA, visceral fat area; SFA, superficial fat area; SMA, skeletal muscle area.

**Figure 2 cancers-13-03598-f002:**
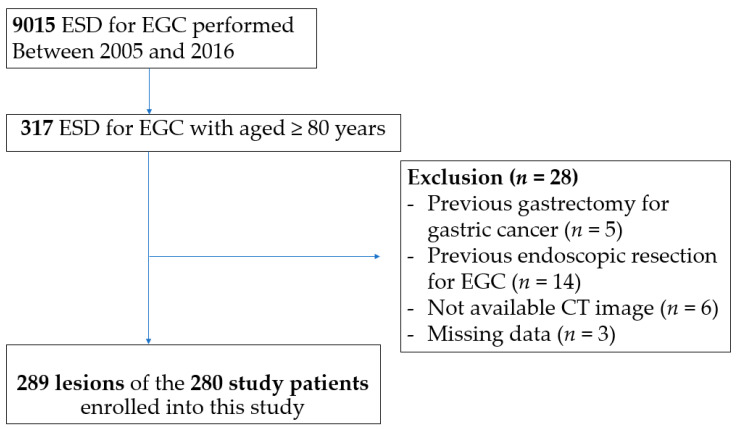
Flow chart of study patient selection. ESD, endoscopic submucosal dissection; EGC, early gastric cancer.

**Figure 3 cancers-13-03598-f003:**
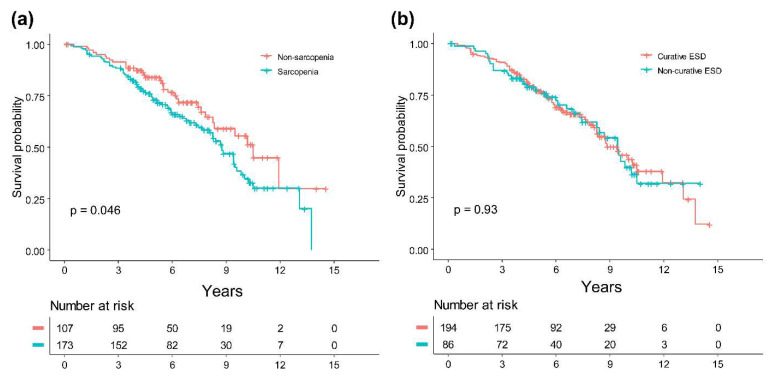
Kaplan–Meier curves for overall survival according to the presence of sarcopenia (**a**) and between those who received curative ESD and those who received non-curative ESD (**b**). OS, overall survival; ESD, endoscopic submucosal dissection.

**Table 1 cancers-13-03598-t001:** Baseline clinical characteristics of the study patients.

Variables	Total (*n* = 280)
Age (years), median (range)	82 (80–92)
Male sex, *n* (%)	182 (65)
ECOG performance status, *n* (%)	
0,1,2	214 (76.4)
3,4	66 (23.6)
Comorbidity (with overlap) *n* (%)	
Hypertension	170 (60.7)
Diabetes mellitus	49 (17.5)
Cardiac disease	49 (17.5)
Cerebrovascular disease	17 (6.1)
Liver cirrhosis	5 (1.8)
Chronic kidney disease	7 (2.5)
Respiratory disease	28 (10)
Malignancy in other organs	23 (8.2)
ASA class, *n* (%)	
1,2	188 (67.1)
3,4, or above	92 (32.9)
Charlson comorbidity index, *n* (%)	
0	132 (47.1)
1	56 (20.0)
2	56 (20.0)
3	23 (8.7)
4	7 (2.5)
5 or above	6 (2.4)
Use of antithrombotic agents, *n* (%)	74 (26.4)
Body mass index (mean, ± SD, kg/m^2^)	23.3 ± 3.4
Prognostic nutritional index (mean, ± SD)	46.7 ± 4.8
SMA index (mean, ± SD)	45.6 ± 6.9
Visceral fat area (mean, ± SD, cm^2^)	120.2 ± 70.5
Sarcopenia, *n* (%)	173 (61.8)
Visceral obesity, *n* (%)	142 (50.7)
VFA/SMA index ratio, median (range)	2.6 (0.2–6.3)
Sarcopenic obesity, *n* (%)	106 (37.9)

Abbreviations: *n*, number; ECOG, Eastern Cooperative Oncology Group; ASA, American Society of Anesthesiologists physical status; SD, standard deviation; SMA, skeletal muscle area; VFA, visceral fat area.

**Table 2 cancers-13-03598-t002:** Characteristics of the 289 lesions in 280 study patients.

Variables	Total (*n* = 289)
Tumor size (mm), median (range)	20 (3–85)
Location, *n* (%)	
Lower	199 (68.9)
Middle	58 (20.1)
Upper	32 (11.1)
Morphology of tumor, *n* (%)	
Elevated	108 (37.4)
Flat	40 (13.8)
Depressed	141 (48.8)
Histologic type, *n* (%)	
Differentiated	263 (91.0)
Undifferentiated	26 (9.0)
Depth of tumor, *n* (%)	
Mucosa	214 (74.0)
Submucosa, SM1	41 (14.2)
Submucosa, SM2	34 (11.8)
Presence of ulcer findings, *n* (%)	19 (6.6)
Lymphovascular invasion, *n* (%)	30 (10.4)
Indication criteria, *n* (%)	
Absolute indication	118 (40.8)
Expanded indication	88 (30.4)
Out of indication	83 (28.7)

Abbreviations: *n*, number, SD; standard deviation; SM1, superficial portion of the submucosa within 500 μm from the muscularis mucosa; SM2 deep portion of the submucosa ≥ 500 μm from the muscularis mucosa.

**Table 3 cancers-13-03598-t003:** Short-term outcomes of 289 lesions (285 ESD sessions) of the 280 study patients.

Variables	Total (*n* = 280)
Procedure time (min), median (range)	25 (4–180)
En bloc resection, *n/N* (%)	287/289 (99.3)
R0 resection, *n/N* (%)	260/289 (90.0)
Curative resection, *n/N* (%)	200/289 (69.2)
Cause of non-curative resection (with overlap), *n/N* (%)	
Positive lateral margin	16/89 (18.0)
Positive deep margin	13/89 (14.6)
Deep submucosal invasion	33/89 (37.1)
Lymphovascular invasion	30/89 (33.7)
Piecemeal resection	1/89 (0.3)
Adverse events of 285 ESD sessions, *n/N* (%)	
Post-ESD Bleeding	17/285 (5.9)
Perforation	6/285 (2.1)
Fever	23/285 (8.1)
Pneumonia	1/285 (0.3)
Stricture	4/285 (1.4)
Treatment-related death	0
Hospital stay (day), median (range)	3 (2–15)
Additional surgery in patients with/ non-curative resection, *n/N* (%)	6/89 (5.6)

Abbreviations: ESD, endoscopic submucosal dissection; n, number; SD, standard deviation.

**Table 4 cancers-13-03598-t004:** Long-term outcomes following endoscopic submucosal dissection.

Variables	Total (*n* = 280)
Follow-up period (months), median (range)	70.5 (4–174)
Local recurrence, *n*/*N* (%)	1/280 (0.3)
Metachronous recurrence, *n*/*N* (%)	19/280 (6.8)
Extragastric recurrence, *n*/*N* (%)	4/280 (1.4)
With intragastric lesion	2/280 (0.7)
Without intragastric lesion	2/280 (0.7)
Number of deaths, *n*/*N* (%)	114/280 (40.7)
Cause of death, *n*/*N* (%)	
Gastric cancer	4/114 (3.5)
Other	110/114 (96.5)

Abbreviations: *n*, number.

**Table 5 cancers-13-03598-t005:** Prognostic factors for survival in univariate and multivariable analyses.

	Univariate		Multivariable	
Variables	HR (95% CI)	*p* Value	HR (95% CI)	*p* Value
Age	1.07 (1.0–1.15)	0.052	1.05 (0.97–1.12)	0.241
Male, sex	0.79 (0.52–1.19)	0.261		
ECOG performance status				
0–1	1			
2 or above	1.4 (0.91–2.15)	0.122		
CCI	1.23 (1.06–1.41)	0.005	1.19 (1.03–1.37)	0.018
PNI	0.93 (0.89–0.97)	<0.001	0.93 (0.90–0.98)	0.002
BMI	0.99 (0.93–1.04)	0.632		
Use of antithrombotic agents	1.15 (0.75–1.76)	0.511		
Tumor size	1.0 (0.99–1.02)	0.516		
Location		0.13		
Lower	1			
Mid	1.28 (0.81–2.04)	0.289		
Upper	1.81 (1.01–3.23)	0.045		
Depth of invasion		0.175		
Mucosa	1			
SM1	1.45 (0.90–2.34)	0.13		
SM2	0.77 (0.43–1.40)	0.394		
Indication of ESD		0.862		
Absolute	1			
Expanded	0.90 (0.57–1.41)	0.624		
Out of indication	1.0 (0.64–1.56)	0.875		
Differentiation				
Differentiated	1			
Undifferentiated	0.94 (0.48–1.86)	0.86		
Lymphovascular invasion				
Absent	1			
Present	1.29 (0.75–2.23)	0.359		
Curability				
Curative resection	1			
Non-curative resection	1.02 (0.68–1.52)	0.934		
Sarcopenia	1.51 (1.01–2.27)	0.048	1.27 (0.84–1.92)	0.266
Visceral obesity	0.97 (0.67–1.40)	0.854		
Sarcopenic obesity	0.99 (0.67–1.45)	0.887		

Abbreviations: HR, hazard ratio; CI, confidence interval; ECOG, Eastern Cooperative Oncology Group; CCI, Charlson comorbidity index; PNI, prognostic nutritional index; BMI, body mass index; SM1, superficial portion of the submucosa within 500 μm from the muscularis mucosa; SM2 deep portion of the submucosa ≥ 500 μm from the muscularis mucosa.

**Table 6 cancers-13-03598-t006:** Clinical and endoscopic factors of the patients who died due to extragastric recurrence of the gastric cancer.

Case No	Sex/ Age	Lesion Size (mm)	Differentiation	Tumor Depth	Resection Margin	LVI	Indication	Sarcopenia	Recurrence				
									Delay (Months)	Site	Characteristics	Management	Survival after Detection (Months)
1	F/85	60	M/D	SM2	R0	Yes	Out of indication	No	66	Lymph node, liver, peritoneum	Extragastric w/o intragastric lesion	Gastrectomy with LND RT	37
2	F/81	15	W/D	SM2	R0	No	Out of indication	No	36	Lymph node, stomach	Extragastric with intragastric lesion	Supportive care	37
3	F/81	25	M/D	SM2	R0	Yes	Out of indication	Yes	23	Lymph node	Extragastric w/o intragastric lesion	Supportive care	3
4	F/86	16	P/D	SM2	R1	No	Out of indication	Yes	38	Lymph node, stomach	Extragastric w/o intragastric lesion	Supportive care	1

Abbreviation: No., number; LVI, lymphovascular invasion; M/D, moderately differentiated; SM2 deep portion of the submucosa ≥ 500 μm from the muscularis mucosa; w/o, without; LND, lymph node dissection; RT, radiation therapy; W/D, well differentiated; P/D, poorly differentiated.

## Data Availability

The data presented in this study are available on request from the corresponding author. The data are not publicly available due to privacy and ethical restrictions.

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
