# Peer review of "Impact of Comorbidities, Sarcopenia, and Nutritional Status on the Long-Term Outcomes after Endoscopic Submucosal Dissection for Early Gastric Cancer in Elderly Patients Aged ≥ 80 Years"

_cancers, 2021, doi:10.3390/cancers13143598_

Round 1

Reviewer 1 Report

This is a retrospective study to evaluate the oncologic outcomes in elderly patients who underwent ESD for early gastric cancer. The study showed that ESD for elderly patients was feasible and safe. Their survival was significantly associated with nutritional status and comorbidity but not sarcopenia judged by skeletal muscle mass using multivariable analysis.

Methods

  1. The authors described that the indication of ESD was based on the Korean Gastric Cancer Association guidelines. However, the rate of curative resection was 69.2%, which seemed that ESD for elderly patients based on guidelines with EGC was over-indicated. I wonder if the authors did not strictly adherent to it.

Results

  1. Thirty-two percent of elderly patients (89/280) were found in non-curative. The author should explain endoscopic diagnosis for ESD indication before treatment.
  2. The most common cause of non-curative resection was deep submucosal invasion (37.1%), and positive vertical margins were observed in 13 (14.6%) tumors. The authors should clarify whether they were overlapped or not. SM massive invasion seemed high. Similar to the above inquiry, the authors should explain it.
  3. Please show the histological findings of 4 patients with extragastric recurrences and deaths. I wondered if the indication of ESD was proper.
  4. Please described the result of local recurrence in the text.
  5. I would like to know the detail of patients (about 10%) with an early poor prognosis (less than three years) because I wonder if they do not have to undergo ESD in case a poor prognosis is predictable. EGC cannot generally cause death within 1 to 2 years even though they were observed without treatment.

Discussion

  1. Once again, I feel discomfort that the author described that more than a quarter of the cases were out of indication for ESD. Does this mean that most of the cases included out of indication for ESD before treatment? If so, this makes the readers confused because the interaction of results can change.
  2. I understand that this is the first report to evaluate the oncological impact on body composition with CT images for elderly patients who underwent ESD for EGC and essential. However, most readers want to know more specific information of what elderly patients do not have to undergo ESD based on these results. If possible, please discuss it or make a recommendation.

Author Response

We thank you and the reviewers for the comments and suggestions on our manuscript entitled, “Impact of comorbidities, sarcopenia and nutritional status on the long-term outcomes after endoscopic submucosal dissection for early gastric cancer in elderly patients aged 80 years” (manuscript ID: cancers-1270126).

 We have carefully read all of the reviewers’ comments, made every effort to address the concerns raised, and applied the corresponding revisions to the manuscript.

Our detailed responses to the reviewer comments are given below in a point-by-point manner, and the corresponding revisions are marked in red, bold letters in the revised manuscript. We hope that our revised manuscript is now acceptable for publication as an Original Article in Cancers.

Response to Reviewer 1 Comments

This is a retrospective study to evaluate the oncologic outcomes in elderly patients who underwent ESD for early gastric cancer. The study showed that ESD for elderly patients was feasible and safe. Their survival was significantly associated with nutritional status and comorbidity but not sarcopenia judged by skeletal muscle mass using multivariable analysis.

Methods

Point 1: The authors described that the indication of ESD was based on the Korean Gastric Cancer Association guidelines. However, the rate of curative resection was 69.2%, which seemed that ESD for elderly patients based on guidelines with EGC was over-indicated. I wonder if the authors did not strictly adherent to it.

Response 1: Thank you for reviewing our study in detail and providing helpful comments. We completely agree with the reviewer’s comment. The rate of curative resection in our study (69.2%) is lower than that in nonelderly patients. Since elderly patients have unique age-related variations in their physical condition, the therapeutic approach for EGC should be differentiated between elderly and nonelderly patients. Surgery carries relatively high rates of morbidity and mortality for elderly patients. Therefore, elderly patients’ general condition should be carefully assessed considering the adverse events associated with general anesthesia and surgery. For this reason, we did not strictly apply the indications for ESD in our study, and included some patients that were out of indication who had poor general condition and severe comorbidities. Based on the updated 2018 Japanese Gastric Cancer Association guidelines (5th edition), endoscopic resection could be an option for the elderly and high-operative-risk patients with severe comorbidities. Such cases are considered as a relative indication and endoscopic resection could be performed, provided that the patient provides consent after being informed of the risk of residual disease.

Results

Point 2: Thirty-two percent of elderly patients (89/280) were found in non-curative. The author should explain endoscopic diagnosis for ESD indication before treatment.

Response 2: Thank you for the careful review and suggestion. We reviewed the medical record again to evaluate the indications before ESD that were found in non-curative resection (89/280). For preprocedure evaluation of disease localization and tumor staging, endoscopy and computed tomography (CT) were performed in almost all cases; however, endoscopic ultrasound (EUS) was performed in 21.3% (19/89) of patients with non-curative resection; among them, deep submucosal invasion was suspected in 25.9% (7/19).

Additionally, we evaluated gross endoscopic findings of the patients with non-curative ESD before procedure. We categorized indication by gross finding and pathologic report of the biopsy specimen. Based on the Korean Gastric Cancer Association guidelines, 33 (37.1%) lesions met the absolute indication and 31 (34.8%) lesions met the expanded indication.

(1) differentiated, nonulcerated, intramucosal, >2 cm in 16; (2) differentiated, ulcerated, intramucosal, ≤3 cm in 2; and (3) undifferentiated, nonulcerated, intramucosal, ≤2 cm in 11 cases) (4) ≤3 cm, predominantly differentiated type, cT1b (SM1) on EUS in 2. ESD was performed in 25 (28.1%) lesions that were out of indication; in these patients, those who were deemed unfit for surgery due to their general condition and high risk of post operative mortality and patient preference were judged to be candidates for ESD.

Point 3: The most common cause of non-curative resection was deep submucosal invasion (37.1%), and positive vertical margins were observed in 13 (14.6%) tumors. The authors should clarify whether they were overlapped or not. SM massive invasion seemed high. Similar to the above inquiry, the authors should explain it.

Response 3: Thank you for the helpful comment. Among the 13 (13/89, 14.6%) cases with vertical margin, 8 cases (8/13, 62%) revealed positive margin that overlapped deep submucosal invasion. We added a sentence on the revised Result section. “Among the 13 (13/89, 14.6%) cases with vertical margin, 8 (8/13, 62%) cases revealed positive margin that overlapped deep submucosal invasion.

Point 4: Please show the histological findings of 4 patients with extragastric recurrences and deaths. I wondered if the indication of ESD was proper.

Response 4: In our study, the overall rate of extragastric recurrence after ESD of EGC was 1.4% (n = 4). Three patients had differentiated histology and one patient had undifferentiated histology. Two patients revealed lymphovascular invasion. All of the four patients with extragastric recurrence were in the out-of-indication group. The decision for undergoing ESD in these patients was made on an individual basis taking into account the patient preference, comorbidities, and performance status. The clinical and endoscopic characteristics of the patients with extragastric recurrence are presented in Table 6, and we corrected the title of Table 6 as follows: “Clinical and endoscopic factors of the patients who died due to extragastric recurrence of the gastric cancer.”

Point 5: Please described the result of local recurrence in the text.

Response 5: In our study, one patient was found to have local recurrence after ESD with a positive lateral margin. This patient was treated by additional argon plasma coagulation and redo ESD. However, the patient underwent gastrectomy after redo ESD because of a positive lateral margin. The patient was alive during the study period after surgery. According to the reviewer’s comment, we added the following sentence in the revised Result section. “One patient was found to have local recurrence after ESD with a positive lateral margin. This patient was treated by additional argon plasma coagulation and redo ESD. However, the patient underwent gastrectomy after redo ESD because of the positive lateral margin. The patient was alive during the study period after surgery.”

Point 6: I would like to know the detail of patients (about 10%) with an early poor prognosis (less than three years) because I wonder if they do not have to undergo ESD in case a poor prognosis is predictable. EGC cannot generally cause death within 1 to 2 years even though they were observed without treatment.

Response 6: We appreciate the reviewer for pointing out this important issue. An adequate answer to this question would require a study comparing these patients with those who received endoscopic treatment and those who were followed without treatment. However, as we mentioned in the Discussion section, we were unable to make a comparison on this issue in the present study. In total, 114 (40.7%, 114/280) patients died during the study period, among whom 29 (25.4%, 29/114) patients had an early poor prognosis (less than three years). Gastric cancer-related death was reported in only one patient with non-curative resection in the follow-up period. The major causes of death were other malignancy (32.1%, 9 cases), pneumonia (10.3%, 3 cases), and cerebrovascular disease (6.9%, 2 cases). At the time of endoscopic treatment, these patients were had good general condition and the diseases were controlled well even if they were accompanied by malignancy in other organs. We believe that further studies and follow-up studies are needed to establish a treatment plan for patients with an early poor prognosis.

Discussion

Point 7: Once again, I feel discomfort that the author described that more than a quarter of the cases were out of indication for ESD. Does this mean that most of the cases included out of indication for ESD before treatment? If so, this makes the readers confused because the interaction of results can change.

Response 7: Thank you for the keen comment. In total, 89 (30.8%) cases were found to be non-curative. Among them, 29 (32.6%, 29/89) cases underwent ESD while acknowledging that the lesion was out of indication. These patients had poor general condition, a high risk of postoperative complication, and expressed preference for ESD over surgery. The decision for gastrectomy was discussed with patients and made on an individual basis by taking into account patient preference and comorbidities. According to the reviewer’s comment, we added the following sentence in the revised Discussion section. “In total, 89 (30.8%) cases were found to be non-curative. Among them, 29 (32.6%, 29/89) cases underwent ESD while acknowledging that the lesion was out of indication. These patients had poor general condition, high risk of postoperative complication, and expressed preference for ESD over surgery. The decision of performing ESD rather than gastrectomy was discussed with patients and made on an individual basis by taking into account patient preference and comorbidities.”

Point 8: I understand that this is the first report to evaluate the oncological impact on body composition with CT images for elderly patients who underwent ESD for EGC and essential. However, most readers want to know more specific information of what elderly patients do not have to undergo ESD based on these results. If possible, please discuss it or make a recommendation.

Response 8: Thank you again for your meticulous review and comments. When we first started this study, we tried to find specific factors that indicated whether or not to perform ESD in elderly patients. However, establishing the cutoff values of CCI or PNI for classifying patients into good/bad candidates for ESD was not one of our planned analysis. We actually explored the relationship between CCI/PNI categories and the outcome using 3-category or 2-category variables and found some signals. However, we believe that testing the difference in the outcomes across the categories on the same dataset used for finding the cutoff values and making inferences about who should receive ESD is rather premature at this stage of the investigation and increases the rate of type I error. Our analysis was aimed to identify risk factors associated with the outcome, and as such, our study can recommend that the decision to perform endoscopic resection in elderly patients should be made with caution when the patient has multiple comorbidities or poor nutritional status. We added the following sentence in the revised Discussion section. “Therefore, the results of our study suggest that the decision to perform endoscopic resection in elderly patients should be made with caution when the patient has multiple comorbidities or poor nutritional status.

Thank you again for reviewing our study and providing critically helpful comments. We hope that our responses and the corresponding revisions are satisfactory.

Reviewer 2 Report

Major comments

The authors claimed to report, in a large population of 280 patients aged over 80 and undergoing endoscopic resection of an early gastric cancer, the impact of the nutritional status on patient outcome. Unfortunately, they did not actually study the technical or oncological outcomes of the endoscopic resection according to the nutritional status, but only the overall survival. The impact of impaired nutritional status or sarcopenia on overall survival is foreseeable, and lacks novelty.

Please consider either studying the impact of nutritional status and sarcopenia on short term outcomes, such as complications after ESD, or on oncological outcomes, such as cancer recurrence; or leave the nutritional data out, and focus on the feasibility, safety and efficacy of gastric ESD in an elderly population.

The data presented suggest a possible extension of the curative criteria for gastric ESD in elderly people, since the patients die from non-cancer related causes.  This is extremely valuable for clinical practice and should be stressed throughout the abstract, discussion, and conclusion.

Please also provide details on the 4 patients with extragastric tumor recurrence.

Minor comments

The definition of perforation on radiography or CT is not consensual, especially since free air/CO2 can be found after the procedure in many cases without perforation. Please use a definition including the visualization a breach in the muscularis propria at endoscopy, fluid in the peritoneum, with or without clinical symptoms.

Among perforations, please distinguish per procedural “perforation” and late onset perforation, those that actually change patient management and usually require surgery.

Page 6, 6/89 patients does not seem to be 5.6 % of the patients. Please amend.

Tables: please define the abbreviations (e.g., “CCI”) used in  the legends

Author Response

We thank you and the reviewers for the comments and suggestions on our manuscript entitled, “Impact of comorbidities, sarcopenia and nutritional status on the long-term outcomes after endoscopic submucosal dissection for early gastric cancer in elderly patients aged 80 years” (manuscript ID: cancers-1270126).

We have carefully read all of the reviewers’ comments, made every effort to address the concerns raised, and applied the corresponding revisions to the manuscript.

Our detailed responses to the reviewer comments are given below in a point-by-point manner, and the corresponding revisions are marked in red, bold letters in the revised manuscript.

Major comments

The authors claimed to report, in a large population of 280 patients aged over 80 and undergoing endoscopic resection of an early gastric cancer, the impact of the nutritional status on patient outcome. Unfortunately, they did not actually study the technical or oncological outcomes of the endoscopic resection according to the nutritional status, but only the overall survival. The impact of impaired nutritional status or sarcopenia on overall survival is foreseeable, and lacks novelty.

Point 1: Please consider either studying the impact of nutritional status and sarcopenia on short term outcomes, such as complications after ESD, or on oncological outcomes, such as cancer recurrence; or leave the nutritional data out, and focus on the feasibility, safety and efficacy of gastric ESD in an elderly population.

Response 1: Thank you for reviewing our study in detail and providing helpful comments. We performed further analysis as you recommended and analyzed the impact of nutritional status and sarcopenia on complications after ESD, cancer recurrence. However, we did not find a significant difference as shown below in Table for reviewer 1,2. In our study, ESD was shown as a feasible and safe therapeutic method in elderly patients. The overall rates of post-ESD bleeding and perforation rates were 5.9% and 2.1%, without significant life-threatening adverse events.

Table for reviewer 1. Association between factors and complication after ESD

   Hazard ratio (95% CI) P-value
Nutritional status    
   PNI ≤ 44.7 1.42 (0.79–2.73) 0.22
   PNI > 44.7 1  
Sarcopenia    
   Sarcopenia 0.98 (0.53–1.81) 0.94
   Non-sarcopenia 1  

CI, confidence interval; PNI, prognostic nutritional index

Table for reviewer 2. Association between factors and recurrence

   Hazard ratio (95% CI) P-value
Nutritional status    
   PNI ≤ 44.7 2.12 (0.96–4.68) 0.064
   PNI > 44.7 1  
Sarcopenia    
   Sarcopenia 1.82 (0.83–3.98) 0.14
   Non-sarcopenia 1  

CI, confidence interval; PNI, prognostic nutritional index

Point 2: The data presented suggest a possible extension of the curative criteria for gastric ESD in elderly people, since the patients die from non-cancer related causes. This is extremely valuable for clinical practice and should be stressed throughout the abstract, discussion, and conclusion.

Response 2: Thank you for your helpful comments to improve our manuscript.

According to the reviewer’s comment, we added the following sentence in the revised Abstract and the conclusion paragraph of the Discussion section. “These results suggest the need for a possible extension of the curative criteria for ESD in elderly patients with EGC.”

We also added the following sentence in the revised Discussion section. “In our study, 114 (40.7%, 114/280) patients died during the study period. Gastric cancer-related death was reported in only four (3.5%, 4/114) patients with non-curative resection in the follow-up period. The remaining 110 patients (96.5%, 110/114) died from-non gastric cancer causes such as other malignancy (32.1%, 9 cases), pneumonia (10.3%, 3 cases), and cerebrovascular disease.”

Point 3: Please also provide details on the 4 patients with extragastric tumor recurrence.

Response 3: In our study, the overall rate of extragastric recurrence after ESD of EGC was 1.4% (n = 4). Three patients had differentiated histology and one patient had undifferentiated histology. Two patients revealed lymphovascular invasion. All of the four patients with extragastric recurrence were in the out-of-indication group. The decision for undergoing ESD in these patients was made on an individual basis taking into account the patient preference, comorbidities, and performance status. The clinical and endoscopic characteristics of the patients with extragastric recurrence are presented in Table 6, and we corrected the title of Table 6 as follows: “Clinical and endoscopic factors of the patients who died due to extragastric recurrence of the gastric cancer.”

Minor comments

Point 4: The definition of perforation on radiography or CT is not consensual, especially since free air/CO2 can be found after the procedure in many cases without perforation. Please use a definition including the visualization a breach in the muscularis propria at endoscopy, fluid in the peritoneum, with or without clinical symptoms.

Response 4: We agree with the reviewer’s comment that the definition of perforation includes the visualization of the breach in the muscularis propria at endoscopy, fluid in the peritoneum, with or without clinical symptoms. In our study, perforation that was according to this definition occurred in 6 procedures. According to the reviewer’s suggestion, we corrected the sentence “Perforation was diagnosed endoscopically or by the presence of free air on either a plain abdominal radiograph or a computed tomography (CT) image.” to “Perforation was defined as the visualization a breach in the muscularis propria at endoscopy, fluid in the peritoneum, with or without clinical symptoms.” in the revised Method section. Additionally, we corrected “perforation rates were 3.1%” to “perforation rates were 2.1%” in the revised abstract and Result section, and “9/285 (3.1)” to “6/285 (2.1)” in the revised Table 3.

Point 5: Among perforations, please distinguish per procedural “perforation” and late onset perforation, those that actually change patient management and usually require surgery.

Response 5: There was no case of late-onset perforation in our study. All perforations were per procedural perforation. We added the following sentences in the revised Result section. “All perforations were per procedural perforation.

Point 6: Page 6, 6/89 patients does not seem to be 5.6 % of the patients. Please amend.

Response 6: We changed the number “5.6” to “6.7” on page 6. Thank you for pointing out our mistake.

Point 7: Tables: please define the abbreviations (e.g., “CCI”) used in the legends

Response 7: We defined the abbreviations in the Table legends.

Thank you again for reviewing our study and providing critically helpful comments. We hope that our responses and the corresponding revisions are satisfactory.

Round 2

Reviewer 1 Report

Thank you for allowing me to review the revised manuscript titled " Impact of comorbidities, sarcopenia and nutritional status on the long-term outcomes after endoscopic submucosal dissection for early gastric cancer in elderly patients aged ≥ 80 years." I have read through the revised manuscript and found that the authors successfully revised the paper and responded to my comments.

Thank you. 

Reviewer 2 Report

The authors adressed the reviewer's comment satisfactorily